# Efficacy and effectiveness of COVID-19 vaccines in Africa: A systematic review

**Tajudeen Raji**[1], **Mosoka Papa Fallah**[1], **Nebiyu Dereje**[1]*, **Francis Kakooza**[2], **Nicaise Ndembi**[1], **Mohammed Abdulaziz**[1], **Merawi Aragaw**[1], **Jean Kaseya**[1], **Alain Ngashi Ngongo**[1]

1 Africa Centres for Disease Control and Prevention (Africa CDC), Addis Ababa, Ethiopia, 2 Infectious Diseases Institute, Makerere University, Kampala, Uganda

☯ These authors contributed equally to this work.

* NebiyuDA@africa-union.org

## Abstract

### Background

Data on COVID-19 vaccine effectiveness to support regional vaccine policy and practice are limited in Africa. Thus, this review aimed to evaluate the efficacy and effectiveness of COVID-19 vaccines administered in Africa.

### Methods

We systematically searched peer-reviewed randomized controlled trials (RCTs), prospective and retrospective cohort studies, and case-control studies that reported on VE in Africa. We carried out a risk of bias assessment, and the findings of this review were synthesized and presented in a narrative form, including tables and figures. The synthesis was focused on COVID-19 VE against various levels of the disease condition and outcomes (infection, hospitalization or critical, and death), time points, and variants of concern.

### Results

A total of 13 studies, with a total sample size of 913,285 participants, were included in this review. The majority (8/13) of studies were from South Africa and 38.5% (5/13) were randomized clinical trials. The studies reported that a full dose of Pfizer-BioNTech vaccine had a VE of 100% against COVID-19 infection by Beta (B.1.351) and Delta variants and 96.7% against hospitalization by Delta variant. The Johnson and Johnson vaccine had VE ranging from 38.1%-62.0% against hospitalization and 51.9%- 86% against critical disease by Beta (B 1.351) variant. The Oxford–AstraZeneca vaccine had a VE of 89.4% against hospitalization by the Omicron variant but was not effective against the B.1.351 variant (10.4%). The Sinopharm vaccine had a VE of 67% against infection and 46% against hospitalization by Delta variant.

### Conclusions

COVID-19 vaccines administered in Africa were effective in preventing infections, hospitalization, and death. These review findings underscore the need for concerted efforts of all

**Data Availability Statement:** All relevant data are within the manuscript and its Supporting Information files.

**Funding:** This research was conducted in partnership with Mastercard Foundation. However,

the authors alone are responsible for the contents of this article and they do not necessarily represent the decisions, policies, or views of Africa CDC or Mastercard Foundation.

**Competing interests:** The authors have declared that no competing interests exist.

stakeholders to enhance the access and availability of COVID-19 vaccines and reinforce public awareness to reach the high-risk, unvaccinated group of the African population.

## Introduction

The COVID-19 pandemic was one of the most devastating public health emergencies that the world faced in the 21[st] century, which claimed the lives of millions of individuals, jeopardized the socio-economic and psychological well-being of the global communities, and severely impacted the health system, particularly in low- and middle-income countries [1, 2]. Fortunately, with the rapid advent and introduction of COVID-19 vaccines, it was possible to mitigate its further impacts–saving lives and livelihoods of the population [3, 4].

However, due to the rapid discovery of the COVID-19 vaccines and the prevailing conspiracies on the quality and safety of the vaccines and misinformation, the vaccine uptake by the African population was limited [5, 6]. As of December 2023, 33% of African populations were vaccinated with a complete primary series of a COVID-19 vaccine and only 6% of the total African populations received booster COVID-19 vaccine. More importantly, as the clinical trials during the vaccine discovery were mainly conducted in developed nations' settings, there was hesitancy about its effectiveness in the African populations [7, 8]. Nevertheless, studies conducted on real-world vaccine effectiveness reflected varying effectiveness of different COVID-19 vaccine types. Different strains of the SARS-COV2 viruses also reflected varying levels of effectiveness [9–11].

In Africa, seven types of COVID-19 vaccines (Johnson & Johnson (Ad26. COV2.S), AstraZeneca (ChAdOx1 nCoV-19 vaccine), Sinopharm vaccine (BBIBP-CorV), Sinovac, Sputnik V, Pfizer-BioNTech (BNT162b2), and Novavax (NVX-CoV2373) vaccines) were introduced and administered to the populations [12]. However, there is a paucity of aggregated data on COVID-19 vaccine effectiveness in African settings. Moreover, a comparative analysis of different vaccine types against variants of concern and various disease levels is imperative to design tailored regional vaccine policy and decision guidance in terms of optimal resource allocation. Evidence on VE might also help in building the capacity of Africa-led science and vaccine discoveries. We, therefore, conducted a systematic review to evaluate the efficacy and effectiveness of COVID-19 vaccines used in Africa and their effectiveness against infection, severity, and death due to SARS-COV2 variants of concern.

## Materials and methods

### Search strategy

We conducted this review as two independent teams; the team from the Makerere University Infectious Diseases Institute and the team from the Africa Centers for Disease Control and Prevention (Africa CDC). We then combined the reviews into this one manuscript. With this approach, we improved the systematic review's reliability, reproducibility, validity, and quality.

We used the Preferred Reporting Items for Systematic Reviews and Meta-Analyses (PRISMA) guideline to conduct this systematic review and report the review findings [13].

We searched MEDLINE (PubMed Central), EMBASE, Web of Science, Cochrane Library, Google Scholar, and direct Google searches for studies published from January 01, 2020 to December 10, 2023. We used keywords, free texts, and Medical Subject Headings (MeSH) terms to search for published articles from the databases. The keywords initially used for our preliminary searches were: COVID-19 vaccine, vaccination, SARS-COV-2 vaccines, effectiveness, efficacy, and Africa. We used the search terms included in the S1 Table to identify studies

from the databases. We also used references from articles to identify additional articles relevant to our study.

## Type of studies

Randomized controlled trials (RCTs), prospective and retrospective cohort studies, longitudinal (follow-up) studies, and case-control studies (including test-negative designs) that reported on vaccine effectiveness in Africa were included in this review.

## Eligibility criteria

We included peer-reviewed publications in the English language that examined the COVID-19 vaccine effectiveness in the African populations. We excluded systematic reviews and meta-analyses, commentaries, reports limited to only the safety of the vaccines, immunogenicity and modelling or simulation studies. Multinational studies that have not reported Africa-specific COVID-19 VE were excluded.

## Intervention (exposure)

COVID-19 vaccines (all types) were considered an exposure status (intervention).

## Control/comparator

The unvaccinated or placebo group of study participants were controls.

## Outcome

The outcome of the review was that vaccine efficacy and effectiveness were assessed and described by the studies in terms of the vaccine's ability to prevent infection and reduce severity (hospitalization) and ICU admission or death. The vaccine efficacy and effectiveness (VE) were expressed by 1—Odds Ratio*100, or 1 –Relative Risk*100, or 1—Hazard Ratio*100, or VE by percentage and 95% confidence interval (CI) or $1 - \frac{risk\ among\ vaccinated}{risk\ among\ unvaccinated} * 100$.

## Quality assessment of studies

The authors determined the risk of bias using the Joanna Briggs Institute (JBI) critical appraisal checklist for RCTs, cohort studies, and case-control (including test-negative design) studies. After the critical appraisal, a score of ≥70% was considered as "low risk", 50–69% was considered "medium risk", and <50% was considered "high risk" [14].

## Data extraction, management and presentation

Based on the data extraction template recommended by the Cochrane Handbook for systematic reviews [15], the authors of the review have extracted data from the full text of the studies on research information (the type of research designs, geographical location of the study, year of publication, sample size of the study, and duration of the follow-up), characteristics of the participants (age, inclusion and exclusion criteria, comorbidity status, SARS-COV-2 strain), intervention (vaccine type, dose, time since administration of the vaccine), control (unvaccinated group characteristics), outcome (how they assessed vaccine effectiveness).

 The authors decided the eligibility of each study to be included in the review by using the eligibility criteria. If any difference occurs about the inclusion of the studies into the review, another external person who is an expert in the area of study was consulted. We used reference managing software (Endnote Version 8.0) to manage data extraction.

## Data synthesis

We conducted a descriptive narrative synthesis of eligible studies included in the review. We summarized the characteristics of the studies by country of origin, study design, vaccine type, and variant of concern. The synthesis was focused on COVID-19 vaccine efficacy and effectiveness against various levels of the disease condition and outcomes (infection, severity (hospitalization) or critical, and death) and variants of concern (e.g., Alpha, Beta, Gamma, Delta, Omicron). The vaccine efficacy and effectiveness against infection were defined as the risk reduction of infection due to SARS-CoV-2 among vaccinated individuals compared to unvaccinated individuals. The vaccine efficacy and effectiveness against hospitalization were determined as the risk reduction of hospitalization among vaccinated individuals compared to unvaccinated individuals. The vaccine efficacy and effectiveness in preventing death was defined as the risk reduction of deaths due to COVID-19 among vaccinated individuals compared to unvaccinated individuals. The review's findings were synthesized and presented in a narrative form, including tables and figures, to aid data presentation where appropriate.

## Results

### Selection and identification of studies

The PRISMA flow diagram (Fig 1) presents the number of studies screened and assessed for eligibility, those included in the synthesis, and the reasons for excluding the studies. Our search brought 908 articles, of which 285 records were duplicates and thus removed. Another 588 articles were excluded after reviewing their title and abstracts. The full texts of the remaining 35 studies were assessed for eligibility, study designs, exposure, and outcome measurements of our interest. Based on this, 22 studies were excluded because they failed to fit with the eligibility criteria. After evaluating each study's score using the quality appraisal criteria, the studies were included in the systematic review. All the studies were scored as low risk of bias (S2 Table).

### Characteristics of the included studies

The 13 studies included in the systematic review had a total sample size of 913,285 and were published from 2020 to 2023. Out of the 13 included studies, 8 (61.5%) of the studies were conducted in South Africa (4 as part of the multinational study, but provided South Africa-specific VE) [16–23], one study (7.1%) were from Egypt [24], and two studies (11.1%) each from Zambia [25, 26] and Morocco [27, 28]. Five studies (42.9%) were randomized clinical trials (RCTs), three studies were case-control (including test-negative design), two studies each were retrospective cohort studies and single-arm implementation trials, and one study was a prospective cohort study. All studies included adult participants aged ≥18 years, except studies by Simwanza J, et al. [26], Thomas S, et al. [17], and Moreira E, et al. [19] who included participants aged ≥13, ≥12, ≥16 years, respectively. There were five types of vaccines included in the studies: Johnson & Johnson (Ad26. COV2.S), AstraZeneca (ChAdOx1 nCoV-19 vaccine, Sinopharm vaccine (BBIBP-CorV), Pfizer-BioNTech, and Novavax (NVX-CoV2373). Studies have reported vaccine effectiveness by various variants of concern, including Omicron, Delta, Alpha (B.1.1.7), Mu (B.1.621), D614G virus, and the Beta (B.1.351) variant (Table 1).

### COVID-19 vaccine efficacy and effectiveness

The studies followed individuals for 7 days– 213 days following the administration of the vaccines to ascertain the vaccine efficacy and effectiveness. The dosage and overall vaccine effectiveness among different populations varies by the vaccine types, time, disease severity, and

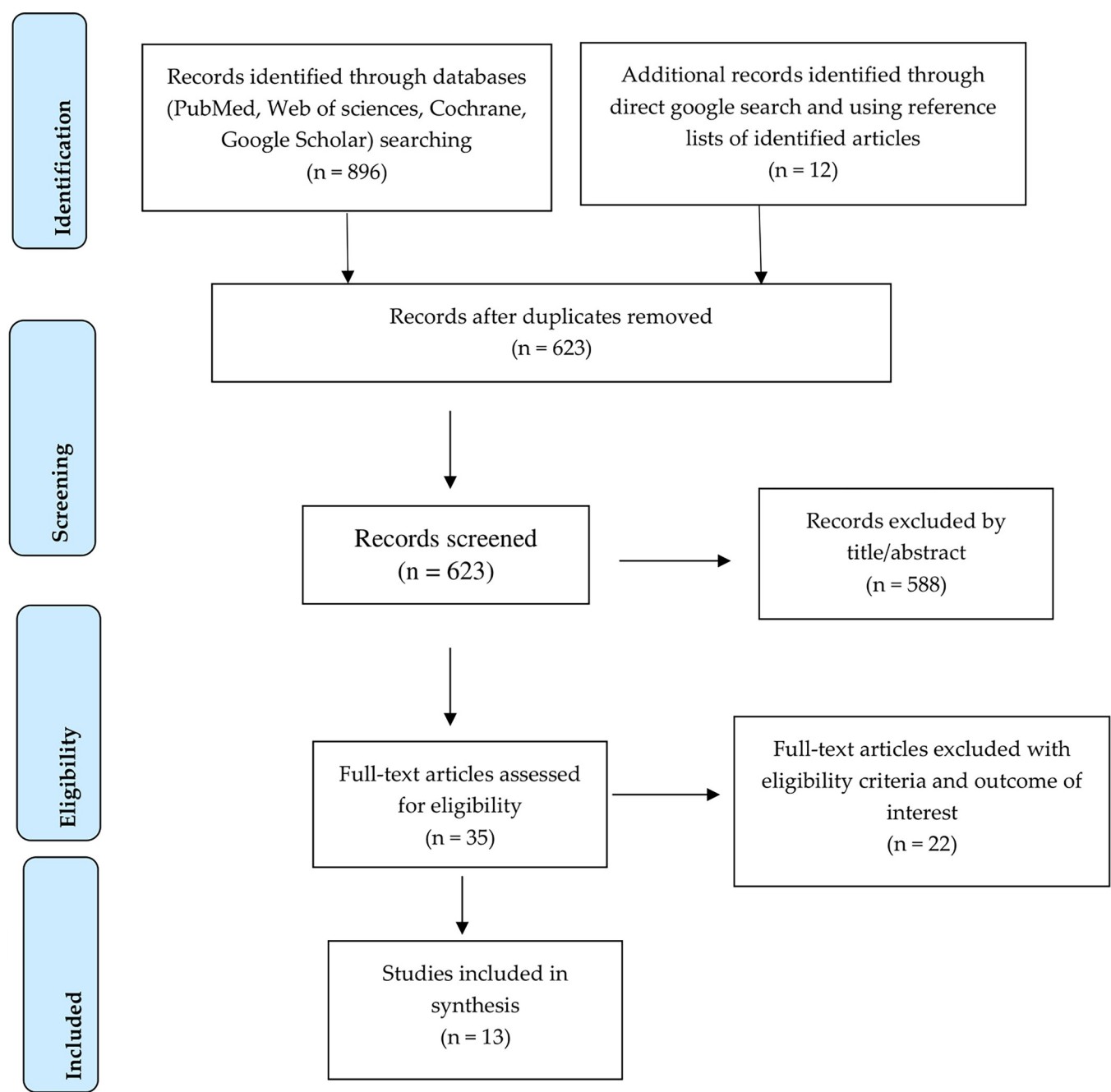

**Fig 1. PRISMA flow diagram showing the screening and identification of studies included in the systematic review.**

variants of concern (Table 2). For example, a study by Gray G, et al. [29] reported 55% (95% CI, 22% -74%) vaccine efficacy against hospitalization within 13 days after the 2nd dose, 74% (95% CI, 57% - 84%) at 14 to 27 days, and 72% (95% CI, 59% - 81%) at 1 to 2 months for Ad26.COV2.S (Johnson and Johnson) vaccine. The Ad26.COV2.S vaccine's efficacy against ICU admission or high care was reported to be 69% (95% CI, 26% - 87%) at 14 to 27 days and 82% (95% CI, 57% - 93%) at 1–2 months after the 2nd dose. The same study reported VE of 81% (95% CI, 41% - 94%) within 13 days after the 2nd dose, 88% (95% CI, 62% - 96%) at 14 to

**Table 1. Characteristics of studies included in the systematic review.**

| Authors | Year | Country | Study design | Sample size in Africa | Age in years | SARS-COV-2 strain | COVID-19 vaccine type |
|---|---|---|---|---|---|---|---|
| Shinde V, et al. [21] | 2021 | South Africa | RCT | 4387 | 18–84 | B.1.351 | NVX-CoV2373 |
| Hardt K, et al. [23] | 2022 | Belgium, Brazil, Colombia, France, Germany, the Philippines, South Africa, Spain, the UK, and the USA | RCT | 2072 | ≥18 | Alpha (B.1.1.7) and Mu (B.1.621) | Ad26.COV2.S |
| Sadoff J, et al. [16] | 2022 | USA, South Africa, Brazil, Colombia, Argentina, Peru, Chile, Mexico | RCT | 856 | 18–60 | Beta (B.1.D614G) Alpha (B.1.1.7), Gamma), C.37 (Lambda), and B.1.621 (Mu), Delta | Ad26.COV2.S |
| Moreira E, et al. [19] | 2022 | USA, Brazil, South Africa | RCT | 268 | ≥16 | Beta (B.1.D614G) Alpha (B.1.1.7), Gamma), C.37 (Lambda), and B.1.621 (Mu), Delta | BNT162b2 |
| Madhi S, et al. [22] | 2021 | South Africa | RCT | 2026 | 18–65 | D614G and B.1.351 variant | ChAdOx1 nCoV-19 vaccine (AZD1222) |
| Thomas S, et al. [17] | 2021 | USA, Brazil, South Africa, Argentina, Turkey, Germany | Single-arm implementation study | 800 | ≥12 | B.1.351 | BNT162b2 |
| Chanda D, et al. [25] | 2022 | Zambia | Retrospective cohort | 1653 | 0–99 | Omicron and Delta | Ad26.COV2.S and ChAdOx1 nCoV-19 vaccine (AZD1222) |
| Zhang Y, et al. [27] | 2022 | Morocco | Retrospective cohort | 348190 | 18–99 | All variants | Sinopharm vaccine (BBIBP-CorV) |
| Simwanza J, et al. [26] | 2021 | Zambia | Case-control | 382 | ≥13 | Omicron and Delta | 90% received Ad26.COV2.S, 10% received ChAdOx1 nCoV-19 vaccine (AZD1222) |
| Gray G, et al. [29] | 2022 | South Africa | Case-control | 93,854 | ≥18 | Omicron variant | BNT162b2 and Ad26.COV2.S |
| Ashmawy R, et al. [24] | 2023 | Egypt | Prospective cohort | 1228 | ≥18 | Not specified | Sinopharm vaccine (BBIBP-CorV) |
| Belayachi J, et al. [28] | 2022 | Morocco | Case-control | 25769 | ≥18 | Not specified | Sinopharm vaccine (BBIBP-CorV) |
| Bekker L, et al. [20] | 2022 | South Africa | Single-arm implementation study | 431626 | ≥18 | Beta (B.1.351) and Delta (B.1.617.2) | Ad26.COV2.S |

27 days, 70% (95% CI, 64% - 76%) at 1 to 2 months, 71% (95% CI, 68% - 74%) at 3 to 4 months, and 67% (95% CI, 63% - 71%) at 5 months or longer for the BNT162b2 (Pfizer-BioNTech) vaccine. The BNT162b2 vaccine's effectiveness against ICU admission or critical care was reported to be 70% (95% CI, 56% to 79%) at 1 to 2 months, 73% (95% CI, 67% to 77%) at 3 to 4 months, and 71% (95% CI, 65% to 76%) at 5 months or longer. Both Ad26.COV2.S (Johnson and Johnson) and BNT162b2 (Pfizer-BioNTech) vaccines were equally effective against severe disease caused by the Omicron variant.

A multinational phase-3 RCT by Sadoff J, et al. [16] reported VE of single dose Ad26. COV2.S (Johnson and Johnson) - 27.2% (95% CI, 16.4% - 36.5%) against infection within 28 days after vaccination; 44.0% (95% CI, 22.9% - 59.7%) and 49.3% (95% CI, 26.9% - 65.3%) against hospitalization within 14 days and 28 days after vaccination, respectively; 70.1% (95% CI, 41.8% - 85.7%) and 75.1% (95% CI, 44.9%-90.1%) against COVID-19-related death 14 days and 28 days after vaccination, respectively. Notably, this trial reported that these levels of vaccine efficacy were achieved despite the high prevalence of the 20H/501Y.V2 SARS-Cov-2

**Table 2. COVID-19 vaccine effectiveness in the included studies.**

| Author | Country | COVID-19 vaccine type | Follow up duration | Dose | VE against infection | VE against hospitalization | VE against ICU admission or death |
|---|---|---|---|---|---|---|---|
| Gray G, et al. [29] | South Africa | BNT162b2 (Pfizer-BioNTech) | 42 days—5 months | Two doses | Not reported | 81% (95% CI, 41% - 94%) within 13 days | Not reported |
| | | | | | | 88% (95% CI, 62% - 96%) at 14 to 27 days | Not reported |
| | | | | | | 70% (95% CI, 64% - 76%) at 1 to 2 months | 70% (95% CI, 56%-79%) at 1 to 2 months |
| | | | | | | 71% (95% CI, 68% - 74%) at 3 to 4 months | 73% (95% CI, 67%-77%) at 3 to 4 months |
| | | | | | | 67% (95% CI, 63% - 71%) at 5 months or longer. | 71% (95% CI, 65%-76%) at 5 months or longer |
| Gray G, et al. [29] | South Africa | Ad26.COV2.S (J & J) | 42 days—5 months | Two doses | Not reported | 55% (95% CI, 22% -74%) within 13 days | Not reported |
| | | | | | | 74% (95% CI, 57% - 84%) at 14 to 27 days, | 69% (95% CI, 26% - 87%) at 14 to 27 days |
| | | | | | | 72% (95% CI, 59% - 81%) at 1 to 2 months | 82% (95% CI, 57% to 93%) at 1 to 2 months |
| Shinde V, et al. [21] | South Africa | NVX-CoV2373 | Seven days following 2nd dose | Two doses 21 days apart | 49.4% (95%CI: 6.1% - 72.8%) | Not reported | Not reported |
| Hardt K, et al. [23] | South Africa* | Ad26.COV2.S (J & J) | Median of 36 days | Two doses 2 months apart | Not reported | 60.0% (95% CI -144.5%–96.2%) | Not reported |
| Sadoff J, et al. [16] | South Africa** | Ad26.COV2.S (J & J) | Median of 4 months | Single dose | 27.2% (95% CI, 16.4% - 36.5%) within 28 days | 44.0% (95% CI, 22.9% - 59.7%) and 49.3% (95% CI, 26.9% - 65.3%) 14 days and 28 days after vaccination, respectively | 70.1% (95% CI, 41.8% - 85.7%) and 75.1% (95% CI, 44.9%-90.1%) 14 days and 28 days after vaccination |
| Moreira E, et al. [19] | USA, Brazil, South Africa | BNT162b2 (Pfizer-BioNTech) | Median of 2.5 months | Three dose | 100.0% (95% CI: -54.9% - 100.0%) within 7 days | Not reported | Not reported |
| Ashmawy R, et al. [24] | Egypt | Sinopharm vaccine (BBIBP-CorV) | Median of 213 days | Two doses | 67% (95% CI: 43–80%) | Not reported | Not reported |
| Belayachi J, et al. [28] | Morocco | Sinopharm vaccine (BBIBP-CorV) | Not specified | Two doses | 1st month = 88% (95% CI, 84–91) | Not reported | Not reported |
| | | | | | 2nd month = 87% (95% CI: 83–90) | | |
| | | | | | 3rd month = 75% (95% CI: 67–80) | | |
| | | | | | 4th month = 61% (95% CI: 54–67) | | |
| | | | | | 5th month = 64% (95% CI: 59–69) | | |
| Bekker L, et al. [20] | South Africa | Ad26.COV2.S (J & J) | Not specified | Single dose | Not reported | 67% (62%– 71%) | 75% (69%– 82%) against COVID-19-related hospital admissions requiring critical or intensive care and 83% (95% CI 75%– 89%) against COVID-19-related deaths |

*(Continued)*

**Table 2.** (Continued)

| Author | Country | COVID-19 vaccine type | Follow up duration | Dose | VE against infection | VE against hospitalization | VE against ICU admission or death |
|---|---|---|---|---|---|---|---|
| Madhi S, et al. [22] | South Africa | ChAdOx1 nCoV-19 vaccine (AZD1222) | Not specified | Two doses 28 days apart | 21.9% (95% CI: -49.9% - 59.8%) | Not reported | Not reported |
| | | | | | A two-dose regimen of the ChAdOx1 nCoV-19 vaccine did not show protection against B.1.351 variant (mild-to-moderate Covid-19) | | |
| Thomas S, et al. [17] | South Africa*** | BNT162b2 (Pfizer-BioNTech) | 6 months follow up | Two doses 28 days apart | 100.0% (95% CI: 53.5–100.0%) within 7 days | 96.7% (80.3–99.9%) | Not reported |
| Chanda D, et al. [25] | Zambia | Ad26.COV2.S and ChAdOx1 nCoV-19 vaccine (AZD1222) | Median 52.5 days | One or more doses | Not reported | 64.8% (95% CI, 42.3%–79.4%) | Not reported |
| Zhang Y, et al. [27] | Morocco | Sinopharm vaccine (BBIBP-CorV) | 14 days after 2nd dose | Two doses 28 days apart | Not reported | 90.2% (95%CI: 87.8% - 92.0%) | Not reported |
| Simwanza J, et al. [26] | Zambia | 90% received Ad26.COV2.S, 10% received ChAdOx1 nCoV-19 vaccine (AZD1222) | Median of 54 days | One or more doses | 64.8% | 72.9% | Not reported |

* Belgium, Brazil, Colombia, France, Germany, the Philippines, South Africa, Spain, the UK, and the USA–The finding included is South Africa-specific VE

** USA, South Africa, Brazil, Colombia, Argentina, Peru, Chile, Mexico—The finding included is South Africa-specific VE

*** USA, Brazil, South Africa, Argentina, Turkey, Germany–The finding included is South Africa-specific VE

variant (Beta South African variant) in South Africa [16]. A subsequent study under the same trial by Hardt K, et al. [23] reported that a booster dose of Ad26.COV2.S (Johnson and Johnson) vaccine administered two months after a single primary dose offered a protective efficacy of 75·2% (95% CI 54·6%– 87·3%) against moderate to severe–critical COVID-19 and 100% (32·6%– 100·0%) against severe–critical COVID-19 by at least 14 days after boosting.

The Sisonke trial [20] conducted among healthcare workers in South Africa evaluated the real-world effectiveness of the Ad26.COV2.S (Johnson and Johnson) vaccine and found that it was 83% effective (95% CI: 75% - 89%) in preventing COVID-19-related deaths, 75% effective (95% CI: 69% - 82%) in preventing COVID-19-related admissions to intensive care, and 67% effective (95% CI: 62% - 71%) in preventing COVID-19-related hospitalizations. The study was conducted during a period when the Beta (B.1.351) and Delta (B.1.617.2) SARS-CoV-2 variants were dominant in the South African population. Still, the efficacy of the vaccine remained consistent–62% (95% CI: 42% - 76%) vaccine efficacy against COVID-19-related hospitalization during the Beta variant wave and 67% (95% CI: 62% - 71%) during the Delta variant wave, and 86% against COVID-19 related death during the Beta variant wave and 82% during the Delta variant wave. The vaccine efficacy was maintained in older healthcare workers and those with comorbidities, including HIV infection [20].

A case-control study to evaluate the real-world vaccine effectiveness of the Sinopharm vaccine (BBIBP-CorV) in Morocco by Belayachi J, et al. [28] reported that the vaccine was highly effective in protecting serious SARS-CoV-2 infection (88% in 1st month and 64% in 5th month of administration). A prospective cohort study conducted in Egypt by Ashmawy R, et al. [24] also reported that the Sinopharm vaccine (BBIBP-CorV) was effective in protecting healthcare workers from COVID-19 (VE = 67% (95% CI: 43% - 80%) (Table 3).

**Table 3. Summary of COVID-19 vaccine effectiveness by vaccine types, variants of concern and disease severity in Africa.**

| Vaccine type | Variants of concern | VE against infection (%) | VE against hospitalization | VE against ICU admission (death) |
|---|---|---|---|---|
| Ad26.COV2.S VE (Johnson and Johnson) | Beta (B 1.351) | Not reported | Varies from 38.1% to 72.0% | Varies from 51.9% to 86% |
| | Alpha (B.1.1.7) | 94.2% | Not reported | Not reported |
| | Mu (B.1.621) | 63.1% | Not reported | Not reported |
| | Delta | Not reported | 67.0% | Varies from 61.1% to 82.0% |
| | Omicron | Not reported | Not reported | 61.1% |
| BNT162b2 (Pfizer-BioNTech) | Beta (B.1.351) | 100.0% | Not reported | Not reported |
| | Delta | 100.0% | Not reported | Not reported |
| | Omicron | Not reported | 88% | Not reported |
| NVX-CoV2373 | B.1.351 | 43.0% | Not reported | Not reported |
| Oxford–AstraZeneca vaccine (ChAdOx1/nCoV-19) | Beta (B.1.351) | 10.4% | Not reported | Not reported |
| | Delta | Not reported | Not reported | 65.2% |
| | Omicron | Not reported | 89.4% | Varies from 65.2% to 85.1% |
| Sinopharm vaccine (BBIBP-CorV) | Delta | 67% | 46% | Not reported |

## Vaccine efficacy and effectiveness by vaccine types and variants of concern

In the included studies, COVID-19 vaccine efficacy and effectiveness vary by the vaccine types, time, and variants of concern (Table 3). The studies included in this review reported Ad26.COV2.S VE (Johnson and Johnson) vaccine effectiveness ranges from 38.1% to 62.0% against hospitalization and 51.9% to 86% against critical disease (ICU admission) for a Beta (B 1.351) variant SARS-COV-2.

## Discussion

COVID-19 vaccines played pivotal roles in the efforts to mitigate the pandemic. With the rapid advent and introduction of the COVID-19 vaccines, several millions of lives were saved, the psycho-social and economic well-being of the people was restored, and the health system was brought back to normal [4, 10]. However, challenges due to emerging and re-emerging variants of concern (VOC) and the decline of vaccine effectiveness over time (vaccine waning) have posed difficulties in vaccination programs [30, 31]. In Africa, where resources are limited and the health literacy level of the populations is minimal, COVID-19 vaccines might not optimally bring the desired changes. Therefore, we conducted a systematic review to evaluate the effectiveness of COVID-19 vaccines against infection, hospitalization (severity), and death.

The principal findings of this systematic review revealed that the full doses of the vaccines administered in Africa were effective against the major variants of concern of SARS-COV-2. A full dose of the BNT162b2 (Pfizer-BioNTech) vaccine was found to have an effectiveness of 100% against COVID-19 infection by Beta (B.1.351) and Delta variants and 96.7% against hospitalization due to Delta variant. The Ad26.COV2.S (Johnson and Johnson) vaccine had a VE of 94.2% against infection caused by the Alpha (B.1.1.7) variant of SARS-COV-2 and 38.1% to 72.0% VE against hospitalization by B.1.351 variant of SARS-COV-2. The Oxford–AstraZeneca vaccine (ChAdOx1/nCoV-19) had a VE of 89.4% against hospitalization by the Omicron variant of SARS-COV-2. However, the Oxford–AstraZeneca (ChAdOx1/nCoV-19) vaccine was not effective against the B.1.351 variant (10.4%). The Sinopharm vaccine (BBIBP-CorV) had a VE of 67% against infection and 46% against hospitalization by Delta variant. The disparities in VE by vaccine types and variants of concern have implications for the identification

of the vaccine type, that is most effective for the populations of Africa, and the appropriate allocation of resources.

Vaccine effectiveness studies, however, may be affected by various biases that may affect their findings [32, 33]. Observational studies are prone to confounding bias, although studies often address this concern. Biases in VE studies can also be introduced during outcome ascertainment and selection of participants (e.g., case-control or cohort studies). Differential testing methods among vaccinated and unvaccinated individuals could also contribute to the biased estimate of VE. Differential depletion of susceptibles–when the vaccine is effective, the people who are infected are more likely to be unvaccinated than vaccinated–could down-estimate the VE. The effect of treatments and passive prophylaxis on VE was not optimally assessed and might have effects on VE. Hence, the findings of the VE studies should be interpreted with caution–having in mind these biases that can be introduced to the studies.

The effectiveness of the vaccines increases with an increase in the recommended vaccine dosage; better prevention in $2^{nd}$ (complete primary series dose) and booster doses can be achieved than in the $1^{st}$ dose [34, 35]. Moreover, vaccines are more effective against severe and critical diseases than mild or asymptomatic infections—for instance, Johnson and Johnson Ad26.COV2.S vaccine effectiveness against moderate to severe–critical COVID-19 was 56.3%, 74.6% against severe–critical COVID-19, and 82.8% against COVID-19-related death. Therefore, enhancing public awareness to continue taking 2nd and booster doses to benefit from the vaccines optimally is critical.

Time after vaccination is also an important parameter reported to influence vaccine effectiveness. As the time interval increases, the vaccine efficacy decreases, and the probability of acquiring infection or worsened severity would be high [36, 37]. A study conducted in Morocco indicated that the Sinopharm vaccine would provide high and stable protection during the first three months (VE of 75%-88%) and decrease after the fourth month (VE of 64%) [27]. Time after vaccination can be lengthened partly by the limited availability of vaccines in the vaccination centres or partly due to the patient's unwillingness to receive further dose(s). Notably, it is important to comply with the vaccination intervals recommended by each vaccine developer, and the general public needs to be aware of it.

This systematic review has some limitations. Two of the multinational studies, which have recruited participants from South Africa, were excluded from this study as they did not report South Africa-specific VE. Articles only published in English language were included in this study and we might have missed studies published in other languages. The risk of bias tool might not optimally capture vaccine effectiveness biases that might be introduced in the specific studies included in this review. Most of the studies were concentrated in a few countries (South Africa and Egypt) and might not represent the actual situation in the entire continent. Most RCTs were conducted in the earlier phases of the COVID-19 pandemic and might not adequately reflect VE for the Omicron variant. Therefore, we recommend further real-world vaccine effectiveness studies in Africa to address this gap.

## Conclusions

This systematic review has revealed the high vaccine effectiveness of COVID-19 vaccines administered in Africa, particularly the commonly used Ad26.COV2.S VE (Johnson and Johnson) vaccine and Oxford–AstraZeneca vaccine (ChAdOx1/nCoV-19) vaccines. These review findings underscore the need for concerted efforts of all stakeholders to enhance the access and availability of COVID-19 vaccines and promote public awareness to reach the high-risk, unvaccinated group of the African populations. Further real-world COVID-19 vaccine effectiveness studies are also required in African settings.

## Supporting information

**S1 Checklist. PRISMA 2020 checklist.**
(DOCX)

**S1 Table. Database search strategy.**
(DOCX)

**S2 Table. Summary of quality assessments using JBI appraisal checklist.**
(DOCX)

## Acknowledgments

The authors are grateful to the authors of the studies included in this systematic review.

## Author Contributions

**Conceptualization:** Tajudeen Raji, Mosoka Papa Fallah, Nebiyu Dereje, Alain Ngashi Ngongo.

**Data curation:** Tajudeen Raji, Mosoka Papa Fallah, Nebiyu Dereje, Francis Kakooza, Nicaise Ndembi, Mohammed Abdulaziz, Merawi Aragaw, Jean Kaseya, Alain Ngashi Ngongo.

**Formal analysis:** Nebiyu Dereje, Francis Kakooza.

**Funding acquisition:** Tajudeen Raji, Mosoka Papa Fallah, Mohammed Abdulaziz, Merawi Aragaw, Jean Kaseya, Alain Ngashi Ngongo.

**Investigation:** Mosoka Papa Fallah, Nebiyu Dereje, Francis Kakooza, Nicaise Ndembi, Mohammed Abdulaziz, Merawi Aragaw, Jean Kaseya, Alain Ngashi Ngongo.

**Methodology:** Nebiyu Dereje, Francis Kakooza, Nicaise Ndembi.

**Project administration:** Tajudeen Raji, Mosoka Papa Fallah, Mohammed Abdulaziz, Merawi Aragaw, Jean Kaseya, Alain Ngashi Ngongo.

**Resources:** Tajudeen Raji, Mosoka Papa Fallah, Mohammed Abdulaziz, Merawi Aragaw, Jean Kaseya, Alain Ngashi Ngongo.

**Software:** Nebiyu Dereje.

**Supervision:** Tajudeen Raji, Mosoka Papa Fallah, Francis Kakooza, Nicaise Ndembi, Mohammed Abdulaziz, Merawi Aragaw, Jean Kaseya, Alain Ngashi Ngongo.

**Validation:** Tajudeen Raji, Mosoka Papa Fallah, Nebiyu Dereje, Francis Kakooza, Nicaise Ndembi, Mohammed Abdulaziz, Merawi Aragaw, Jean Kaseya, Alain Ngashi Ngongo.

**Visualization:** Nebiyu Dereje, Nicaise Ndembi, Mohammed Abdulaziz, Merawi Aragaw, Jean Kaseya.

**Writing – original draft:** Tajudeen Raji, Mosoka Papa Fallah, Nebiyu Dereje, Francis Kakooza, Nicaise Ndembi.

**Writing – review & editing:** Tajudeen Raji, Mosoka Papa Fallah, Nebiyu Dereje, Francis Kakooza, Nicaise Ndembi, Mohammed Abdulaziz, Merawi Aragaw, Jean Kaseya, Alain Ngashi Ngongo.

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
