## [Decision Letter · Decision Letter 0]

7 May 2024

PONE-D-24-04033Efficacy and effectiveness of COVID-19 vaccines in AfricaPLOS ONE

Dear Dr. Dereje,

Thank you for submitting your manuscript to PLOS ONE. After careful consideration, we feel that it has merit but does not fully meet PLOS ONE’s publication criteria as it currently stands. Therefore, we invite you to submit a revised version of the manuscript that addresses the points raised during the review process.

We look forward to receiving your revised manuscript.

Kind regards,

Ashraful Hoque,

Academic Editor

PLOS ONE

Journal Requirements:

2. During your revisions, please revise your title to specify that your study is a Systematic Review, and update it in the manuscript file and online submission information if needed.

Reviewers' comments:

Reviewer's Responses to Questions

**Comments to the Author**

1. Is the manuscript technically sound, and do the data support the conclusions?

Reviewer #1: Partly

Reviewer #2: Partly

2. Has the statistical analysis been performed appropriately and rigorously? 

Reviewer #1: N/A

Reviewer #2: Yes

3. Have the authors made all data underlying the findings in their manuscript fully available?

Reviewer #1: Yes

Reviewer #2: Yes

4. Is the manuscript presented in an intelligible fashion and written in standard English?

Reviewer #1: Yes

Reviewer #2: Yes

5. Review Comments to the Author

Reviewer #1: Overall comments:

The authors have crafted a well-written article on an interesting and important topic. Nevertheless, the article would benefit from addressing several issues before publication:

1. The authors highlight the lack of data as a reason that data are needed, but it would be helpful to clarify the rationale more explicitly. a) If there is a question about generalizability of estimates from other regions (in general, VE estimates seem to be generalizable, but one could argue that needs to be demonstrated).b) demonstrating effectiveness at a regional level might be critical to support vaccine policy c) it might help in building regional capacity for African-led science. It would help to differentiate these issues more explicitly.

2. The article could be improved by synthesizing data across studies for different estimates. The results section reads as a narrative summary of different articles, but it would be clearer to organize the results by what these estimates refer to. For example, what was the effectiveness of a primary series between 7 days and 3 months against symptomatic infection, or against hospitalization? Time since vaccination is particularly important to make explicit because VE can vary dramatically by time since dose, particularly against milder outcomes.

- Instead of summarizing ‘overall VE’ in the abstract, it is important to specify the endpoint(s) (symptomatic infection? Hospitalization? Severe illness?) and product (primary series 7 days to 6 months? mRNA within specified time period). Otherwise, it’s unclear whether the wide range in estimates reflects inclusion of different types of VE. For example, one would expect much lower VE if defined as against mild illness any time after vaccination, compared with VE against severe illness within 3 months of vaccination. Of course, variant is also worth including, but it would be okay to pool estimates for pre-Omicron variants (or pre-Delta and Delta) if needed to simplify the summary of overall estimates.

- In Table 3, the right column is difficult to follow. Instead, recommend presenting estimates for specific endpoints and products, each on a different line. As an example of an effective way to do this, please see the PDF tables in the IVAC review of COVID-19 VE studies (see https://view-hub.org/vaccine/covid/effectiveness-studies?field_region=african)

- As an example of the need to avoid comparing ‘apples and oranges’ in the abstract, it is not clear why estimates presented for the Astra-Zeneca vaccine are somewhat higher than the Pfizer-BioNTech vaccine. This seems at odds with the literature, and likely reflects differences in the studies cited for each. To address this, as above it would be clearer to ensure that each estimate has a clear endpoint and time since dose.

- By clarifying the different endpoints and vaccines, the authors could provide greater clarity on gaps in the evidence for African populations – which endpoints and vaccines did or didn’t have any estimates? It it also important to clarify the referent group – for example, vaccinated individuals may represent the referent group for estimation of the benefit of a third dose.

3. The rationale is unclear for including studies that do not provide region-specific estimates. It is an important point that large multi-country trials included populations in Africa. However, if no regional estimates were reported, they are outside scope of assessing effectiveness in Africa. It is fine for the authors to note this distinction, but it would be clearer if the estimates are limited to VE in Africa. Overall, recommend limiting the analysis to studies that produced estimates of efficacy or VE in African populations.

4. It would be clearer to report efficacy studies separately from observational studies (after considering whether they report any Africa-specific estimates). For example, the multicountry trials might not have any such estimates, whereas the trial by Madhi et al would be good to include. It is not clear why immunogenicity studies would be included. The cross-sectional survey of staff at an Egyptian University does not appear to have any estimates of VE and could also be excluded.

5. In the discussion, it would be helpful for the authors to clarify the extent to which variation in estimates might reflect different bias in different studies. In general, it would be helpful first to consider whether variation reflects different products or endpoints, then different study biases, before considering potential variability between populations. It is striking from the VE literature that even though there may be some differences by age and underlying health conditions, estimates are generally similar by these characteristics and across demographic groups.

6. Some data appears to be missing in the tables (for example, the number of adults in Sadoff et al). If this is unknown, it would be better to specify. If the study doses not report estimates limited to Africa, would omit this from the review (see above). In Table 2, it’s not clear why age is presented differently (sometimes ‘adults’, sometimes an age range, median age, ‘All’). Recommend completing the tables consistently.

7. In the discussion, it would be important to highlight the challenge of biases in VE studies (see the summary on WHO’s website (see https://www.who.int/publications/i/item/WHO-2019-nCoV-vaccine_effectiveness-VE_evaluations-2022.1 ). This is important as a caveat to interpretation of some specific estimates, and might explain some of the variability between studies, even when comparing the same products, timing, and endpoints.

8. Recommend focusing on VE or efficacy by recommended primary series or additional doses. Would report VE of 1st doses only as a secondary and separate point (given that this might be considered to partially taking a recommended product, even though some coverage is better than none). This could be made clearer in the discussion (line 154).

Abstract:

L12-13: Recommend rewording to clarify why estimates are needed for Africa. For example, could highlight few studies, and challenges in coverage in Africa. Also please specify this refers to COVID-19 vaccines.

L14: If you need to save words, there is no need to list each database in the abstract.

Background:

General: Recommend clarify the different reasons to address this questin, e.g.: 1) Although VE estimates might be broadly generalizable, important to demonstrate in Africa 2) Important to help inform regional policy and help improve coverage. 3) Important that Africa has this capacity – to be able to monitor effectiveness of new interventions.

L57: Consider referring to as African ‘populations’ rather than one ‘population’.

L60: Please specify the endpoints that VE is assessed against (e.g., any infection, symptomatic infection..)

Methods:

L75: ?typo (“effect”)

L76: It would be easier for the reader to move the search term to the supplement and just reference it here. You might also be able to rely on Table 1 instead, which could also be moved to the supplement.

L91: From the figure, it looks like the references from articles were also used to find additional articles. Can add a sentence here to clarify the additional search method.

L96: Note that there there is some nuance about the definition of a test-negative design, which can generally be considered to be a subset of the case-control design (see https://pubmed.ncbi.nlm.nih.gov/31430265/) . Some studies might be labeled ‘test-negative’ but allow participants to know there case-control status at enrollment. For simplicity, recommend simply referring to ‘case-control’ throughout. A sentence can be added to explain that ‘test-negative’ was considered a subset of case-control studies.

L99: Would be curious about whether limitation to English language publications might have missed some reports in Francophone Africa / other countries. Recommend adding a line in the limitations to acknowledge this.

L109: Several endpoints are listed here – it would be better to specify if several were assessed. E.g., ‘VE or efficacy were assessed against several endpoints: symptomatic infection, hospitalization,…’

L114: See comment above about the test-negative design, which might be considered a subset of ‘case-control’. Recommend clarifying how ‘test-negative’ was classified, and which tools were used for which design.

L124: Consider summarizing by time since dose and specifying that here – since VE wanes over time, it is important to consider limitations from comparing estimates with substantially different times since last vaccine dose.

L125: What does ‘proportion protected’ mean here?

L129: Would omit the statement about duplications, since it should be a given that you did not report the same study twice. Okay to include in the results flow chart though.

L135: Recommend consistent capitalization of SARS-CoV-2 variants.

L141: Would not define VE or efficacy as ‘% reduction’ compared to unvaccinated individuals. For example, if deaths were decreased from 200/100,000 to 50/100,000 among the vaccinated, and remained 200 per 100,000 among the unvaccinated, VE would be 1-(0.5/2)= 75%.

Results:

L176: Efficacy or VE was not estimated at 7 days, correct? Suggest rephrasing this as during the period of 7 days to 213 days. This range highlights a challenge in trying to summarize overall estimates.

L179: Rephrase ‘concern variants’

Tables and Figure:

General: Would limit articles presented to those with efficacy or VE estimates limited to African populations.

Table 2: Would present only articles with estimates for populations within Africa

Table 3: There is a lot of content in the right column, which would be better presented as separate lines for each endpoint. Separate lines could also be added for each product, and time since dose. Recommend specifying ‘estimated’ VE or efficacy in the column title.

Table 4: Would recommend parsing out the content where it represents a wide range of variants or time since dose. For example, the range of VE from the Pfizer-BioNTech vaccine is reported as 52%-100% but it’s not clear from the table if this reflects variation between studies, endpoints, variants, or time since dose. Instead, this row could be sub-divided into rows for pre-Omicron and Omicron, by endpoint (symptomatic infection, severe illness) or similar.

Discussion:

L244: Would mention challenged of waning VE (could be argued to be more important than variant per se) and also limited coverage.

L268: Which guidelines are referred to as ‘it’? Recommend rewording and citing as appropriate. Would also soften this language as it’s hard to argue that the results presented make it ‘imperative’ to follow the unspecified recommendations.

L270: Recommend omitting studies that did not provide estimates of VE or efficacy for African populations.

L270ff: Add the limitation that the risk of bias tool might not have identified specific biases that could still have substantial effects on VE estimates (and consider citing summaries of bias in VE studies)

References:

Recommend ensuring that references are cited in the order in which they appear, using appropriate reference manager software.

Also, the reference 21 refers to an immunogenicity study, whereas the efficacy estimate in the tables appears to be from reference 19. It would be good to correct this, and double-check if any of the other references are misplaced.

In the table, references are sometimes given by the first name (e.g., ‘John S’). Recommend using last names instead, per usual convention.

Reviewer #2: PONE-D-24-04033

Efficacy and effectiveness of COVID-19 vaccines in Africa

Major comments:

1. This review assumes that the results of COVID-19 vaccine RCTs conducted in one part of the world might not apply to other parts, in particular Africa.

a. Has this been true for other vaccine RCTs, eg, influenza, measles, etc?

b. Are the Africa-based VE estimates summarized in this review *substantially different* from VE estimates derived from RCTs conducted in other parts of the world eg US/Europe? If so, that would add to the evidence that these kinds of geographically distinct RCTs and systematic reviews are necessary.

2. Agree with the authors who note in the Discussion section that “most of the studies were concentrated in a few countries (South Africa and Egypt) and might not represent the actual situation in the entire continent.” This major weakness undermines much of the point of the whole review. A stronger counterargument would help here.

3. Also agree with their statement that “most RCTs were conducted in the earlier phases of the COVID-19 pandemic and might not adequately reflect VE for the omicron variant.” So one is left wondering how to apply the findings from this review to the current situation.

4. Beyond RCTs, ‘real-world’ test-negative designs might be more important than RCTs in terms of comparing VE in different parts of the world. This review found only two such studies. I agree with the authors’ concluding recommendation of “further real-world vaccine effectiveness studies in Africa.”

5. The review offers relatively little in the way of synthesizing, interpretive commentary. Instead, it serves more as a comprehensive source that lists out all in one place the COVID-19 vaccine trials that were conducted in Africa.

6. There is no comment at all about the impact of HIV status on VE, nor other immunocompromising conditions. This is important since most of the studies were conducted in South Africa, where HIV prevalence is about 14%.

7. Funding: on page 1, the authors state “The author(s) received no specific funding for this work.” Yet later in the paper they describe that MasterCard funded the study. Better to make this all clear and consistent.

Methods:

Were the bibliographies of reviews and meta analyses scanned for additional studies?

Results:

“Another 588 articles [of 623] were excluded after reviewing their title and abstracts.” This means 94% of the identified and deduplicated articles were excluded without much detail as to why.

Minor:

Language is not always clear and correct. A copyedit would help with, e.g., tense, as past and present tense are both used throughout.

Some authors are referred to by their first names eg Glenda G and Linda GB

Table S1, page 2, “S John et al” row is blank for the risk assessment

I suggest the authors consider adding as context a brief description of actual COVID-19 vaccine uptake in Africa.

6. PLOS authors have the option to publish the peer review history of their article (what does this mean?). If published, this will include your full peer review and any attached files.

Reviewer #1: No

Reviewer #2: No

---

## [Author Response · Author response to Decision Letter 0]

13 Jun 2024

Dear Dr. Ashraful Hoque,

Academic Editor

PLOS ONE,

Thank you for the opportunity to respond to the editor’s and reviewers’ comments. We

appreciate their insightful comments and suggestions. We now have revised the manuscript accordingly and uploaded the revised version to the system. Please kindly find the point-by-point responses to each comment provided below. 

Formatting requirements

Response: Thank you. We have formatted the manuscript now according to PLOS ONE’s style.

2. During your revisions, please revise your title to specify that your study is a Systematic Review, and update it in the manuscript file and online submission information if needed.

Response: Thank you. We include now in the title that the study is a systematic review.

Response: We have corrected the funding information accordingly.

Reviewer 1 comments and responses

Overall comment: The authors have crafted a well-written article on an interesting and important topic. Nevertheless, the article would benefit from addressing several issues before publication:

Response: Thank you very much for the insightful comments and suggestions that have improved our manuscript. 

1. The authors highlight the lack of data as a reason that data are needed, but it would be helpful to clarify the rationale more explicitly. a) If there is a question about the generalizability of estimates from other regions (in general, VE estimates seem to be generalizable, but one could argue that needs to be demonstrated).b) demonstrating effectiveness at a regional level might be critical to support vaccine policy c) it might help in building regional capacity for African-led science. It would help to differentiate these issues more explicitly.

Response: Thank you for this comment. The three rationales indicated by the reviewer are used to justify the need for this systematic review.

2. The article could be improved by synthesizing data across studies for different estimates. The results section reads as a narrative summary of different articles, but it would be clearer to organize the results by what these estimates refer to. For example, what was the effectiveness of a primary series between 7 days and 3 months against symptomatic infection, or against hospitalization? Time since vaccination is particularly important to make explicit because VE can vary dramatically by time since dose, particularly against milder outcomes.

Response: Thank you for this insightful comment. We now have revised to include vaccine effectiveness for different time points and levels of protection (against infection, against hospitalization, and against ICU admission/death). 

- Instead of summarizing ‘overall VE’ in the abstract, it is important to specify the endpoint(s) (symptomatic infection? Hospitalization? Severe illness?) and product (primary series 7 days to 6 months? mRNA within specified time period). Otherwise, it’s unclear whether the wide range in estimates reflects inclusion of different types of VE. For example, one would expect much lower VE if defined as against mild illness any time after vaccination, compared with VE against severe illness within 3 months of vaccination. Of course, variant is also worth including, but it would be okay to pool estimates for pre-Omicron variants (or pre-Delta and Delta) if needed to simplify the summary of overall estimates.

Response: Thank you. We now revise the abstract accordingly to include specific VE by time, product, and variants of concern. We have also provided the details of this in the body of the manuscript.

- In Table 3, the right column is difficult to follow. Instead, recommend presenting estimates for specific endpoints and products, each on a different line. As an example of an effective way to do this, please see the PDF tables in the IVAC review of COVID-19 VE studies (see https://view-hub.org/vaccine/covid/effectiveness-studies?field_region=african)

- As an example of the need to avoid comparing ‘apples and oranges’ in the abstract, it is not clear why estimates presented for the Astra-Zeneca vaccine are somewhat higher than the Pfizer-BioNTech vaccine. This seems at odds with the literature, and likely reflects differences in the studies cited for each. To address this, as above it would be clearer to ensure that each estimate has a clear endpoint and time since dose.

- By clarifying the different endpoints and vaccines, the authors could provide greater clarity on gaps in the evidence for African populations – which endpoints and vaccines did or didn’t have any estimates? It it also important to clarify the referent group – for example, vaccinated individuals may represent the referent group for estimation of the benefit of a third dose.

Response: Thank you very much for this critical comment. We have revised Table 3 accordingly to provide VE by different times (endpoints) and levels of disease prevention. The Table now is improved and provides clearer information.

3. The rationale is unclear for including studies that do not provide region-specific estimates. It is an important point that large multi-country trials included populations in Africa. However, if no regional estimates were reported, they are outside scope of assessing effectiveness in Africa. It is fine for the authors to note this distinction, but it would be clearer if the estimates are limited to VE in Africa. Overall, recommend limiting the analysis to studies that produced estimates of efficacy or VE in African populations.

Response: Only two of the multi-national studies did not provide country-specific VE and were excluded from the review as suggested. Only those multi-national studies with Africa-specific estimates are included in this synthesis.

4. It would be clearer to report efficacy studies separately from observational studies (after considering whether they report any Africa-specific estimates). For example, the multicountry trials might not have any such estimates, whereas the trial by Madhi et al would be good to include. It is not clear why immunogenicity studies would be included. The cross-sectional survey of staff at an Egyptian University does not appear to have any estimates of VE and could also be excluded.

Response: Thank you. We separately reported for RCTs and observational studies. We also removed the cross-sectional survey and immunogenicity study as they do not directly estimate and report VE.

5. In the discussion, it would be helpful for the authors to clarify the extent to which variation in estimates might reflect different bias in different studies. In general, it would be helpful first to consider whether variation reflects different products or endpoints, then different study biases, before considering potential variability between populations. It is striking from the VE literature that even though there may be some differences by age and underlying health conditions, estimates are generally similar by these characteristics and across demographic groups.

Response: Thank you. We discuss the biases that might have contributed to the variations of VE in the included studies. 

6. Some data appears to be missing in the tables (for example, the number of adults in Sadoff et al). If this is unknown, it would be better to specify. If the study doses not report estimates limited to Africa, would omit this from the review (see above). In Table 2, it’s not clear why age is presented differently (sometimes ‘adults’, sometimes an age range, median age, ‘All’). Recommend completing the tables consistently.

Response: Thank you. We now include Africa-specific sample size and estimate for the Sadoff et al. in Table 2. We now revise the age of participants in Table 2 to complete it consistently.

7. In the discussion, it would be important to highlight the challenge of biases in VE studies (see the summary on WHO’s website (see https://www.who.int/publications/i/item/WHO-2019-nCoV-vaccine_effectiveness-VE_evaluations-2022.1 ). This is important as a caveat to interpretation of some specific estimates, and might explain some of the variability between studies, even when comparing the same products, timing, and endpoints.

Response: Thank you. We have included the biases in the discussion. 

8. Recommend focusing on VE or efficacy by recommended primary series or additional doses. Would report VE of 1st doses only as a secondary and separate point (given that this might be considered to partially taking a recommended product, even though some coverage is better than none). This could be made clearer in the discussion (line 154).

Response: Thank you. Revised accordingly. 

Abstract:

L12-13: Recommend rewording to clarify why estimates are needed for Africa. For example, could highlight few studies, and challenges in coverage in Africa. Also please specify this refers to COVID-19 vaccines.

Response: Thank you. Revised accordingly.

L14: If you need to save words, there is no need to list each database in the abstract.

Response: Thank you. Revised accordingly.

Background:

General: Recommend clarify the different reasons to address this questin, e.g.: 1) Although VE estimates might be broadly generalizable, important to demonstrate in Africa 2) Important to help inform regional policy and help improve coverage. 3) Important that Africa has this capacity – to be able to monitor effectiveness of new interventions.

Response: Thank you. We now add reasons to conduct the review.

L57: Consider referring to as African ‘populations’ rather than one ‘population’.

Response: Thank you. Revised accordingly.

L60: Please specify the endpoints that VE is assessed against (e.g., any infection, symptomatic infection..)

Response: Thank you. Revised accordingly.

Methods:

L75: ?typo (“effect”)

Response: Thank you. We removed it now.

L76: It would be easier for the reader to move the search term to the supplement and just reference it here. You might also be able to rely on Table 1 instead, which could also be moved to the supplement.

Response: Thank you. We moved now the search strategy and the Table to a supplementary file.

L91: From the figure, it looks like the references from articles were also used to find additional articles. Can add a sentence here to clarify the additional search method.

Response: Yes, we have used references from articles to identify additional articles relevant to our study. We state this in the text now.

L96: Note that there there is some nuance about the definition of a test-negative design, which can generally be considered to be a subset of the case-control design (see https://pubmed.ncbi.nlm.nih.gov/31430265/) . Some studies might be labeled ‘test-negative’ but allow participants to know there case-control status at enrollment. For simplicity, recommend simply referring to ‘case-control’ throughout. A sentence can be added to explain that ‘test-negative’ was considered a subset of case-control studies.

Response: Agree. We now treat test-negative designs as case-control studies and revise accordingly.

L99: Would be curious about whether limitation to English language publications might have missed some reports in Francophone Africa / other countries. Recommend adding a line in the limitations to acknowledge this.

Response: Thank you. We acknowledge this in the limitations.

L109: Several endpoints are listed here – it would be better to specify if several were assessed. E.g., ‘VE or efficacy were assessed against several endpoints: symptomatic infection, hospitalization,…’

Response: Thank you. We now specify the end-points for the estimation of VE. 

L114: See comment above about the test-negative design, which might be considered a subset of ‘case-control’. Recommend clarifying how ‘test-negative’ was classified, and which tools were used for which design.

Response: Thank you. Revised accordingly.

L124: Consider summarizing by time since dose and specifying that here – since VE wanes over time, it is important to consider limitations from comparing estimates with substantially different times since last vaccine dose.

Response: Thank you. Revised accordingly.

L125: What does ‘proportion protected’ mean here?

Response: It was to mean the proportion of vaccinated individuals protected from acquiring infection. As we removed the cross-sectional study from this review, we do not need to include this statement now. 

L129: Would omit the statement about duplications, since it should be a given that you did not report the same study twice. Okay to include in the results flow chart though.

Response: Agree. Thank you.

L135: Recommend consistent capitalization of SARS-CoV-2 variants.

Response: Thank you. Revised accordingly.

L141: Would not define VE or efficacy as ‘% reduction’ compared to unvaccinated individuals. For example, if deaths were decreased from 200/100,000 to 50/100,000 among the vaccinated, and remained 200 per 100,000 among the unvaccinated, VE would be 1-(0.5/2)= 75%.

Response: Thank you. We revised it to ‘risk reduction’. 

Results:

L176: Efficacy or VE was not estimated at 7 days, correct? Suggest rephrasing this as during the period of 7 days to 213 days. This range highlights a challenge in trying to summarize overall estimates.

Response: Thank you. We have rephrased it for clarity. 

L179: Rephrase ‘concern variants’

Response: Corrected. Thank you.

Tables and Figure:

General: Would limit articles presented to those with efficacy or VE estimates limited to African populations.

Response: Agreed and revised accordingly.

Table 2: Would present only articles with estimates for populations within Africa

Response: Agreed and revised accordingly.

Table 3: There is a lot of content in the right column, which would be better presented as separate lines for each endpoint. Separate lines could also be added for each product, and time since dose. Recommend specifying ‘estimated’ VE or efficacy in the column title.

Response: Thank you. Revised Table 3 accordingly.

Table 4: Would recommend parsing out the content where it represents a wide range of variants or time since dose. For example, the range of VE from the Pfizer-BioNTech vaccine is reported as 52%-100% but it’s not clear from the table if this reflects variation between studies, endpoints, variants, or time since dose. Instead, this row could be sub-divided into rows for pre-Omicron and Omicron, by endpoint (symptomatic infection, severe illness) or similar.

Discussion:

L244: Would mention challenged of waning VE (could be argued to be more important than variant per se) and also limited coverage.

Response: Thank you. We added vaccine waning as one of the challenges. 

L268: Which guidelines are referred to as ‘it’? Recommend rewording and citing as appropriate. Would also soften this language as it’s hard to argue that the results presented make it ‘imperative’ to follow the unspecified recommendations.

Response: We are referring to the intervals to receive subsequent doses of COVID-19 vaccines after the primary dose as indicated by the vaccine developers. We have revised the sentence to clarify this.

L270: Recommend omitting studies that did not provide estimates of VE or efficacy for African populations.

Response: Thank you. We have omitted multinational studies that did not provide VE specific to African populations.

L270ff: Add the limitation that the risk of bias tool might not have identified specific biases that could still have substantial effects on VE estimates (and consider citing summaries of bias in VE studies)

References: Recommend ensuring that references are cited in the order in which they appear, using appropriate reference manager software. Also, the reference 21 refers to an immunogenicity study, whereas the efficacy estimate in the tables appears to be from reference 19. It would be good to correct this, and double-check if any of the other references are misplaced. In t

---

## [Editor Report · Acceptance letter]

19 Jun 2024

PONE-D-24-04033R1 

PLOS ONE

Dear Dr. Dereje, 

I'm pleased to inform you that your manuscript has been deemed suitable for publication in PLOS ONE. Congratulations! Your manuscript is now being handed over to our production team.

Kind regards, 

on behalf of

Dr. Ashraful Hoque 

Academic Editor

PLOS ONE